# Position-based Scaled Gradient for Model Quantization and Pruning

**Jangho Kim**
Seoul National University
Seoul, Korea
kjh91@snu.ac.kr

**KiYoon Yoo**
Seoul National University
Seoul, Korea
961230@snu.ac.kr

**Nojun Kwak**[*]
Seoul National University
Seoul, Korea
nojunk@snu.ac.kr

## Abstract

We propose the position-based scaled gradient (PSG) that scales the gradient depending on the position of a weight vector to make it more compression-friendly. First, we theoretically show that applying PSG to the standard gradient descent (GD), which is called PSGD, is equivalent to the GD in the warped weight space, a space made by warping the original weight space via an appropriately designed invertible function. Second, we empirically show that PSG acting as a regularizer to the weight vectors is favorable for model compression domains such as quantization and pruning. PSG reduces the gap between the weight distributions of a full-precision model and its compressed counterpart. This enables the versatile deployment of a model either as an uncompressed mode or as a compressed mode depending on the availability of resources. The experimental results on CIFAR-10/100 and ImageNet datasets show the effectiveness of the proposed PSG in both domains of pruning and quantization even for extremely low bits. The code is released in Github[2].

## 1 Introduction

Many regularization strategies have been proposed to induce a prior to neural networks [20, 38, 19, 23]. Inspired by such regularization methods which induce a prior for a specific purpose, in this paper we propose a novel regularization method that non-uniformly scales gradient for model compression problems. The scaled gradient, whose scale depends on the position of the weight, constrains the weight to a set of compression-friendly grid points. We replace the conventional gradient in the stochastic gradient descent (SGD) with the proposed position-based scaled gradient (PSG) and call it as PSGD. We show that applying PSGD in the original weight space is equivalent to optimizing the weights by the standard SGD in a warped space, to which weights from the original space are warped by an invertible function. The invertible warping function is designed such that the weights of the original space are forced to merge to the desired target positions by scaling the gradients.

We are not the first to scale the gradient elements. The *scaled gradient method* which is also known as the *variable metric method* [9] multiplies a positive definite matrix to the gradient vector to scale the gradient. It includes a wide variety of methods such as the Newton method, Quasi-Newton methods and the natural gradient method [11, 34, 4]. Generally, they rely on Hessian estimation or Fisher information matrix for their scaling. However, our method is different from them in that our scaling does not depend on the loss function but it depends solely on the current position of the weight.

We apply the proposed PSG method to the model compression problems such as quantization and pruning. In recent years, deploying a deep neural network (DNN) on restricted edge devices such

---

[*]Corresponding Author
[2]https://github.com/Jangho-Kim/PSG-pytorch

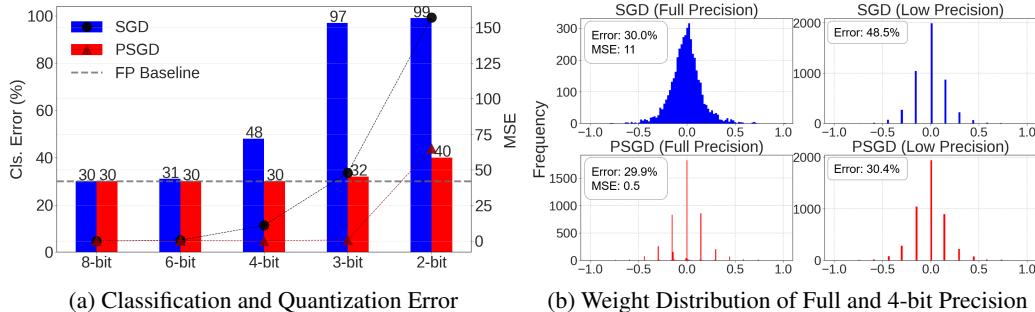

| (a) Classification and Quantization Error | (b) Weight Distribution of Full and 4-bit Precision |

Figure 1: Results of ResNet-34 on CIFAR-100. (a) Mean-squared quantization error (line) and classification error (bar) across different bits. Blue: SGD, Red: PSGD. (b) Example of weight distribution (Conv2_1 layer [18]) trained with standard SGD and our PSGD. For PSGD, the distribution of the full precision weights closely resembles the low precision distribution, yet maintains its accuracy.

as smartphones and IoT devices has become a very important issue. For these reasons, reducing bit-width of model weights (quantization) and removing unimportant model weights (pruning) have been studied and widely used for applications. Majority of the literature in quantization, dubbed as Quantization Aware Training (QAT) methods, fine-tunes a pre-trained model on the low precision domain without considering the full precision domain using the entire training dataset. Moreover, this scenario is restrictive in real-world applications because additional training is needed. In the additional training phase, a full-size dataset and high computational resources are required which prohibits easy and fast deployment of DNNs on edge devices for customers in need.

To resolve this problem, many works have focused on post-training quantization (PTQ) methods that do not require full-scale training [25, 32, 2, 41]. For example, [32] starts with a pre-trained model with only minor modification on the weights by equalizing the scales across channels and correcting biases. However, inherent discrepancy in the distribution of the pre-trained model and that of the quantized model is too large for the aforementioned methods to offset the fundamental difference in the distributions. As shown in Fig. 1, due to the differences in the two distributions, the classification error and the quantization error, denoted as the mean squared error increase as lower bit-width is used. Accordingly, when it comes to layer-wise quantization, existing post-training methods suffer significant accuracy degradation when it is quantized below 6-bit.

Meanwhile, another line of research in quantization has recently emerged that approaches the task from the initial training phase [1]. Our method follows this scheme of training from scratch like standard SGD, but we attain a competent full-precision model that can also be effortlessly quantized to a low precision model with no additional post-processing. In essence, our main goal is to train a compression-friendly model that can be easily compressed when the resources are limited, without the need of re-training, fine-tuning and even accessing the data. To achieve this, we constrain the original weights to merge to a set of quantized grid points (Appendix A and Fig. 1(b)) by scaling their gradients proportional to the error between the original weight and its quantized version. For pruning, the weights are regularized to merge to zero. More details will be described in Sec 3.

Our contributions can be summarized as follows:

• We propose a novel regularization method for model compression by introducing the position-based scaled gradient (PSG) which can be considered as a variant of the variable metric method.

• We prove theoretically that PSG descent (PSGD) is equivalent to applying the standard gradient descent in the warped weight space. This leads the weight to converge to a well-performing local minimum in both compressed and uncompressed weight spaces (see Appendix A and Fig. 1).

• We apply PSG in quantization and pruning and verify the effectiveness of PSG on CIFAR and ImageNet datasets. We also show that PSGD is very effective for extremely low bit quantization. Furthermore, when PSGD-pretrained model is used along with a concurrent PTQ method, it outperforms its SGD-pretrained counterpart.

## 2 Related work

**Quantization**    QAT methods have shown increasingly strong performance in the low-precision domain even to 2,3 bit-width [12, 14, 3, 22]. Post-training quantization, on the other hand, aims to quantize weights and activation without additional training or using the training data. Majority of the works in recent literature starts from a pre-trained network trained by standard training scheme [41, 32, 2]. Many works on channel-wise quantization methods, which require storing quantization parameters per channel, have shown notable improvement in performance even at 4-bit [2, 8]. However, layer-wise quantization methods, which are more hardware-friendly as they store quantization parameters per layer (as opposed to per channel), still suffers at lower bit-widths [32, 25, 41]. [32] achieves near full-precision accuracy at 8-bit by bias correction and range equalization of channels, while [41] splits channels with outliers to reduce the clipping error. However, both suffer from severe accuracy degradation under 6-bit. Our method improves on but is not limited to the uniform layer-wise quantization. Concurrent to ours, [33] and [31] propose to directly minimize the quantization error using a calibration dataset to achieve higher performance at under 6-bit. We show using PSGD pretrained model outperforms using SGD pretrained model in Section 5.

Meanwhile, another line of work in quantization has focused on quantization robustness by regularizing the weight distribution from the initial training phase. [29] focuses on minimizing the Lipshitz constant to regularize the gradients for robustness against adversarial attacks. Similarly, [1] proposes a new regularization term on the norm of the gradients for quantization robustness across different bit widths. This enables "on-the-fly" quantization to various bit widths. Our method does not have an explicit regularization term but scales the gradients to implicitly regularize the weights in the full-precision domain to make them quantization-friendly. Additionally, we do not introduce significant training overhead because gradient norm regularization is not necessary, while [1] necessitates double-backpropagation which increases the training complexity. Some other related works in quantization aims to quantize the gradient vectors for efficient training [10], propose more representative encoding formats [37], or learn the optimal mixed precision bit-width [39].

**Pruning**    Another relevant line of research in model compression is pruning, in which unimportant units such as weights, filters, or entire blocks are pruned [21, 28]. Recent works have focused on pruning methods that include the pruning process in the training phase [35, 42, 30, 27]. Among them, substantial amount of works utilize sparsity-inducing regularization. [30] proposes training with L0 norm regularizer on individual weights to train a sparse network, using the expected L0 objective to relax the otherwise indifferentiable regularization term. Meanwhile, other works focus on using saliency criterion. [27] utilizes gradients of masks as a proxy for importance to prune networks at a single-shot. Similar to [27] and [30], our method does not need a heuristic pruning schedule during training nor additional fine-tuning after pruning. In our method, pruning is formulated as a subclass of quantization because PSG can be used for sparse training by setting the target value as zero instead of the quantized grid points.

## 3 Proposed method

In this section, we describe the proposed position-based scaled gradient descent (PSGD) method. In PSGD, a scaling function regularizes the original weight to merge to one of the desired target points which performs well at both uncompressed and compressed domains. This is equivalent to optimizing via SGD in the warped weight space. With a specially designed invertible function that warps the original weight space, the loss function in this warped space converges to a different local minima that are more compression-friendly compared to the solutions driven in the original weight space.

We first prove that optimizing in the original space with PSGD is equivalent to optimizing in the warped space with gradient descent. Then, we demonstrate how PSGD is used to constrain the weights to a set of desired target points. Lastly, we provide explanation on how this method is able to yield comparable performance with that of vanilla SGD in the original uncompressed domain, despite its strong regularization effect.

### 3.1 Optimization in warped space

**Theorem:** Let $\mathcal{F} : \mathcal{X} \to \mathcal{Y}$, $\mathcal{X}, \mathcal{Y} \subset \mathbb{R}^n$, be an arbitrary invertible multivariate function that warps the original weight space $\mathcal{X}$ into $\mathcal{Y}$ and consider the loss function $\mathcal{L} : \mathcal{X} \to \mathbb{R}$ and the equivalent loss

function $\mathcal{L}' = \mathcal{L} \circ \mathcal{F}^{-1} : \mathcal{Y} \to \mathbb{R}$. Then, the gradient descent (GD) method in the warped space $\mathcal{Y}$ is equivalent to applying a scaled gradient descent in the original space $\mathcal{X}$ such that

$$GD(\boldsymbol{y}, \nabla_{\boldsymbol{y}}^{\mathcal{L}'}) \equiv GD(\boldsymbol{x}, (\mathcal{J}_{\boldsymbol{x}}^{\mathcal{F}})^{-2} \nabla_{\boldsymbol{x}}^{\mathcal{L}}), \tag{1}$$

where $\boldsymbol{y} = \mathcal{F}(\boldsymbol{x})$ and $\nabla_a^b$ and $\mathcal{J}_a^b$ respectively denote the gradient and Jacobian of the function $b$ with respect to the variable $a$.

**Proof:** Consider the point $\boldsymbol{x}_t \in \mathcal{X}$ at time $t$ and its warped version $\boldsymbol{y}_t \in \mathcal{Y}$. To find the local minimum of $\mathcal{L}'(\boldsymbol{y})$, the standard gradient descent method at time step $t$ in the warped space can be applied as follows:

$$\boldsymbol{y}_{t+1} = \boldsymbol{y}_t - \eta \nabla_{\boldsymbol{y}}^{\mathcal{L}'}(\boldsymbol{y}_t). \tag{2}$$

Here, $\nabla_{\boldsymbol{y}}^{\mathcal{L}'}(\boldsymbol{y}_t) = \frac{\partial \mathcal{L}'}{\partial \boldsymbol{y}}|_{\boldsymbol{y}_t}$ is the gradient and $\eta$ is the learning rate. Applying the inverse function $\mathcal{F}^{-1}$ to $\boldsymbol{y}_{t+1}$, we obtain the updated point $\boldsymbol{x}_{t+1}$:

$$\boldsymbol{x}_{t+1} = \mathcal{F}^{-1}(\boldsymbol{y}_{t+1}) = \mathcal{F}^{-1}(\boldsymbol{y}_t - \eta \nabla_{\boldsymbol{y}}^{\mathcal{L}'}(\boldsymbol{y}_t)) = \mathcal{F}^{-1}(\boldsymbol{y}_t) - \eta \mathcal{J}_{\boldsymbol{y}}^{\boldsymbol{x}}(\boldsymbol{y}_t) \nabla_{\boldsymbol{y}}^{\mathcal{L}'}(\boldsymbol{y}_t) \tag{3}$$

where the last equality is from the first-order Taylor approximation around $\boldsymbol{y}_t$ and $\mathcal{J}_{\boldsymbol{y}}^{\boldsymbol{x}} = \mathcal{J}_{\boldsymbol{y}}^{\mathcal{F}^{-1}} = \frac{\partial \boldsymbol{x}}{\partial \boldsymbol{y}} \in \mathbb{R}^{n \times n}$ is the Jacobian of $\boldsymbol{x} = \mathcal{F}^{-1}(\boldsymbol{y})$ with respect to $\boldsymbol{y}$. By the chain rule, $\nabla_{\boldsymbol{y}}^{\mathcal{L}'} = \frac{\partial \boldsymbol{x}}{\partial \boldsymbol{y}} \frac{\partial \mathcal{L}}{\partial \boldsymbol{x}} = \mathcal{J}_{\boldsymbol{y}}^{\boldsymbol{x}} \nabla_{\boldsymbol{x}}^{\mathcal{L}}$. Because $\mathcal{J}_{\boldsymbol{y}}^{\boldsymbol{x}} = (\mathcal{J}_{\boldsymbol{x}}^{\boldsymbol{y}})^{-1} = (\mathcal{J}_{\boldsymbol{x}}^{\mathcal{F}})^{-1}$, we can rewrite Eq.(3) as

$$\boldsymbol{x}_{t+1} = \boldsymbol{x}_t - \eta (\mathcal{J}_{\boldsymbol{x}}^{\mathcal{F}}(\boldsymbol{x}_t))^{-2} \nabla_{\boldsymbol{x}}^{\mathcal{L}}(\boldsymbol{x}_t). \tag{4}$$

Now Eq.(2) and Eq.(4) are equivalent and Eq.(1) is proved. In other words, the scaled gradient descent (PSGD) in the original space $\mathcal{X}$, whose scaling is determined by the matrix $(\mathcal{J}_{\boldsymbol{x}}^{\mathcal{F}})^{-2}$, is equivalent to gradient descent in the warped space $\mathcal{Y}$.

### 3.2 Position-based scaled gradient

In this part, we introduce one example of designing the invertible function $\mathcal{F}(\boldsymbol{x})$ for scaling the gradients. This invertible function should cause the original weight vector $\boldsymbol{x}$ to merge to a set of desired target points $\{\bar{\boldsymbol{x}}\}$. These kinds of desired target weights can act as a prior in the optimization process to constrain the original weights to merge at specific positions. The details of how to set the target points will be deferred to the next subsection.

The gist of weight-dependent gradient scaling is simple. For a given weight vector, if the specific weight element is far from the desired target point, a higher scaling value is applied so as to escape this position faster. On the other hand, if the distance is small, lower scaling value is applied to prevent the weight vector from deviating from the position. From now on, we focus on the design of the scaling function for the quantization problem. For pruning, the procedure is analogous and we omit the detail.

**Scaling function:** We use the same warping function $f$ for each coordinate $x_i, i \in \{1, \cdots, n\}$ independently, i.e. $\boldsymbol{y} = \mathcal{F}(\boldsymbol{x}) = [f(x_1), f(x_2), \cdots f(x_n)]^T$. Thus the Jacobian matrix becomes diagonal ($\mathcal{J}_{\boldsymbol{x}}^{\mathcal{F}} = \mathrm{diag}(f'(x_1), \cdots, f'(x_n))$) and our method belongs to the diagonally scaled gradient method.

Consider the following warping function

$$f(x) = 2\,\mathrm{sign}(x - \bar{x})(\sqrt{|x - \bar{x}| + \epsilon} - \sqrt{\epsilon}) + c(\bar{x}) \tag{5}$$

where the target $\bar{x}$ is determined as the closest grid point from $x$, $\mathrm{sign}(x) \in \{\pm 1, 0\}$ is a sign function and $c(\bar{x})$ is a constant dependent on the specific grid point $\bar{x}$ making the function continuous[3]. $\epsilon$ is an arbitrarily small constant to avoid infinite gradient. Then, from Eq.(4), the elementwise scaling function becomes $s(x) = \frac{1}{[f'(x)]^2}$ and consequently

$$s(x) = |x - \bar{x}(x)| + \epsilon. \tag{6}$$

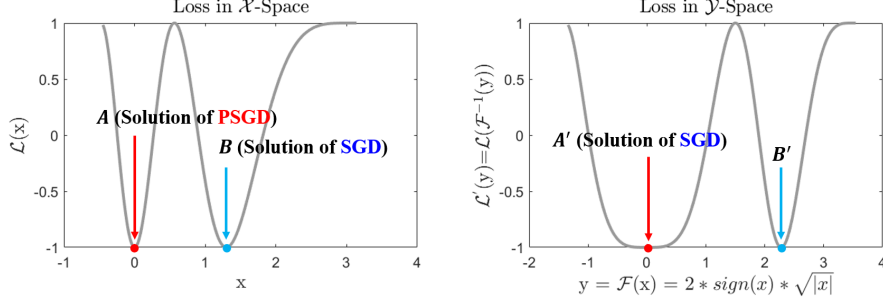

Figure 2: Toy example of warping a loss function $\mathcal{L}(x) = \cos\left((x - 3.07)^2\right)$. **Left** denotes the original loss function. **Right** is drawn by warping the original function by Eq. (5) with the target $\bar{x} = 0$.

Using the elementwise scaling function Eq.(6), the elementwise weight update rule for the PSG descent (PSGD) becomes

$$x_i{}^{t+1} = x_i{}^t - \eta s(x_i) \left.\frac{\partial \mathcal{L}}{\partial x_i}\right|_{\boldsymbol{x}^t} \tag{7}$$

where, $\eta$ is the learning rate[4].

### 3.3 Target points

**Quantization:** In this paper, we use the uniform symmetric quantization method [25] and the per-layer quantization scheme for hardware friendliness. Consider a floating point range $[min_x, max_x]$ of model weights. The weight $x$ is quantized to an integer ranging $[-2^{n-1} + 1, 2^{n-1} - 1]$ for n-bit precision. Quantization-dequantization for the weights of a network is defined with step-size ($\Delta$) and clipping values. The overall quantization process is as follows:

$$x_Q = Clip(\left\lfloor \frac{x}{\Delta} \right\rceil, -2^{n-1} + 1, 2^{n-1} - 1), \quad \Delta = \frac{max(-min_x, max_x)}{2^{n-1} - 1} \tag{8}$$

where $\lfloor \cdot \rceil$ is the round to the closest integer operation and $Clip(x, a, b) = \begin{cases} b & \text{if} \quad x > b \\ a & \text{if} \quad x < a \\ x & \text{elsewise.} \end{cases}$

We can get the quantized weights with the de-quantization process as $\bar{x} = x_Q \times \Delta$ and use this quantized weights for target positions of quantization.
**Pruning:** For magnitude-based pruning methods, weights near zero are removed. Therefore, we choose zero as the target value (i.e. $\bar{x} = 0$).

### 3.4 PSGD for deep networks

Many literature focusing on the optimization of DNNs with stochastic gradient descent (SGD) have reported that multiple experiments give consistently similar performance although DNNs have many local minima (e.g. see Sec. 2 of [6]). [7] analyzed the loss surface of DNNs and showed that large networks have many local minima with similar performance on the test set and the lowest critical values of the random loss function are located in a specific band lower-bounded by the global minimum. From this respect, we explain informally how PSGD for deep networks works. As illustrated in Fig. 2, we posit that there exist many local minima $(A, B)$ in the original weight space $\mathcal{X}$ with similar performance, only some of which $(A)$ are close to one of the target points (0) exhibiting high performance also in the compressed domain. As in Fig. 2 left, assume that the region of convergence for $B$ is much wider than that of $A$, meaning that there exists more chance to output solution $B$ rather than $A$ from random initialization. By the warping function $\mathcal{F}$ specially designed as described above (Eq. 5), the original space $\mathcal{X}$ is warped to $\mathcal{Y}$ such that the areas near target points are expanded while those far from the targets are contracted. If we apply gradient descent in this warped space, the loss function will have a better chance of converging to $A'$. Correspondingly, PSGD in the

Table 1: Test accuracy of ResNet-32 across different sparsity ratios (percentage of zeros) on CIFAR-100 after magnitude-based pruning [17] without any fine-tuning.

| Method | Sparsity (%) | | | | |
|---|---|---|---|---|---|
| | 20.0 | 50.0 | 70.0 | 80.0 | 90.0 |
| SGD | 69.43 | 60.59 | 15.95 | 4.70 | 1.00 |
| L0 Reg. [30] | 67.56 | 64.49 | 49.73 | 23.95 | 2.85 |
| SNIP [27] | **69.68** | 68.73 | 66.76 | 65.67 | 60.14 |
| PSGD (Ours) | 69.63 | **69.25** | **68.62** | **67.27** | **64.33** |

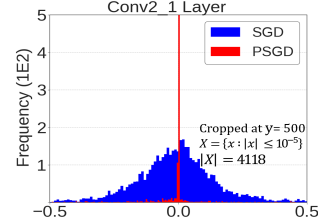

Figure 3: The weight distribution of SGD and PSGD models.

original space will more likely output $A$ rather than $B$, which is favorable for compression. Note that $\mathcal{F}$ transforms the original weight space to the warped space $\mathcal{Y}$ not to the compressed domain.

## 4 Experiments

In this section, we experimentally show the effectiveness of PSGD. To verify our PSGD method, we first conduct experiments for sparse training by setting the target point as 0, then we further extend our method to quantization with CIFAR [26] and ImageNet ILSVRC 2015 [36] dataset. We first demonstrate the effectiveness in sparse training with magnitude-based pruning by comparing with L0-regularization [30] and SNIP [27]. [30] penalizes the non-zero model parameters and shares the scheme of regularizing the model while training. Like ours, [27] is a single-shot pruning method, which does not require pruning schedules nor additional fine-tuning.

For quantization, we compare our method with **(1)** methods that employ regularization at the initial training phase [1, 16, 29]. We choose gradient L1 norm regularization [1] method and Lipschitz regularization methods [29, 16] from the original paper [1] as baselines, because they propose new regularization techniques used at the training phase similar to us. Note that [16] adds an L2 penalty term on the gradient of weights instead of the L1 penalty like [1]. We also compare with **(2)** existing state-of-the-art layer-wise post-training quantization methods that start from pre-trained models [32, 41] to show the improvement in lower bits (4-bit). Refer to Section 2 for the details on the compared methods. To validate the effectiveness of our method, we also train our model for extremely low bit (2,3-bit) weights. Lastly, we show the experimental results on various network architectures and applying PSG to the Adam optimizer [24], which are detailed in Appendix D.

**Implementation details**  We used the Pytorch framework for all experiments. For the pruning experiment of Table 1, we used ResNet-32 [18] on the CIFAR-100, following the training hyper-parameters of [40]. We used released official implementations of [30] and re-implemented [27] for the Pytorch framework. For quantization experiments of Table 2 and 3, we used ResNet-18 and followed [1] settings for CIFAR-10 and ImageNet. For [41], released official implementations were used for experiment. All other numbers are either from the original paper or re-implemented. For fair comparison, all quantization experiments followed the layer-wise uniform symmetric quantization [25] and when quantizing the activation, we clipped the activation range using batch normalization parameters as described in [32], same as [1]. PSGD is applied from the last 15 epochs for ImageNet experiments and from the first learning rate decay epoch for CIFAR experiments. We use additional 30 epochs for PSGD at extremely low bits experiments (Table 4). Also, we tuned the hyper-parameter $\lambda_s$ for each bit-widths and sparsity. Our search criteria is ensuring that the performance of uncompressed model is not degraded, similar to [1]. More details are in Appendix C.

### 4.1 Pruning

As a preliminary experiment, we first demonstrate that PSG-based optimization is possible with a single target point set at zero. Then, we apply magnitude-based pruning following [17] across different sparsity ratios. As the purpose of the experiment is to verify that the weights are centered on zero, weights are pruned once after training has completed and the model is evaluated without fine-tuning for [30] and ours. Results for [27], which prunes the weights by single-shot at initialization, are shown for comparison on single-shot pruning.

Table 2: Test accuracy of regularization methods that do not have post-training process for ResNet-18 on the ImageNet and CIFAR dataset. PSGD@W# indicates the target number of bits for weights in PSGD is #. All numbers except ours are from [1]. At #-bit, PSGD@W# performs the best in most cases.

| Method | ImageNet | | | | CIFAR-10 | | | |
|---|---|---|---|---|---|---|---|---|
| | FP | W8A8 | W6A6 | W4A4 | FP | W8A4 | W6A4 | W4A4 |
| SGD | 69.70 | 69.20 | 63.80 | 0.30 | 93.54 | 85.51 | 85.35 | 83.98 |
| DQ Regularization [29] | 68.28 | 67.76 | 62.31 | 0.24 | 92.46 | 83.31 | 83.34 | 82.47 |
| Gradient L2 [16] | 68.34 | 68.02 | 64.52 | 0.19 | 93.31 | 84.50 | 84.99 | 83.82 |
| Gradient L1 [1] | 70.07 | 69.92 | 66.39 | 0.22 | 93.36 | 88.70 | 88.45 | 87.62 |
| Gradient L1 ($\lambda = 0.05$) [1] | 64.02 | 63.76 | 61.19 | 55.32 | – | – | – | – |
| PSGD@W8 (Ours) | **70.22** | **70.13** | 66.02 | 0.60 | **93.67** | **93.10** | 93.03 | 90.65 |
| PSGD@W6 (Ours) | 70.07 | 69.83 | **69.51** | 0.29 | 93.54 | 92.76 | 92.88 | 90.55 |
| PSGD@W4 (Ours) | 68.18 | 67.63 | 62.73 | **63.45** | 93.63 | 93.04 | **93.12** | **91.03** |

Table 3: Comparison with Post-training Quantization methods using ResNet-18 on the ImageNet dataset. Results of DFQ are from [32].

| Method | W8A8 | W6A6 | W4A4 |
|---|---|---|---|
| DFQ [32] | 69.7 | 66.3 | – |
| OCS + Best Clip [41] | 69.37 | 66.76 | 44.3 |
| PSGD (Ours) | **70.13** | **69.51** | **63.45** |

Table 4: Extremely low bits accuracy of ResNet-18 on the ImageNet dataset. The first convolutional layer and the last linear layer are quantized at 8-bit. Activation is fixed to 8-bit.

| Method | (FP / W3A8) | (FP / W2A8) |
|---|---|---|
| SGD | **69.76** / 0.10 | **69.76** / 0.10 |
| PSGD (Ours) | 66.75 / **66.36** | 64.60 / **62.65** |

Table 1 indicates that our method outperforms the two methods across various high sparsity ratios. While all three methods are able to maintain accuracy at low sparsity ($\sim$10%), [30] has some accuracy degradation at 20% and suffers severely at high sparsity. This is in line with the results shown in [13] that the method was unable to produce sparse residual models without significant damage to the model quality. Comparing with [27], our method is able to maintain higher accuracy even at high sparsity, displaying the strength in single-shot pruning, in which no pruning schedules nor additional training are necessary. Fig. 3 shows the distribution of weights in SGD- and PSGD-trained models.

## 4.2  Quantization

In the quantization domain, we first compare PSGD with regularization methods at the on-the-fly bit-widths problem, meaning that a single model is evaluated across various bit-widths. Then, we compare with existing state-of-the-art layer-wise symmetric post-training methods to verify handling the problem of accuracy drop at low bits due to the differences in weight distributions (See Fig. 1).

**Regularization methods**     Table 2 shows the results of regularization methods on CIFAR-10 and ImageNet datasets, respectively. In the CIFAR-10 experiments of Table 2, we fix the activation bit-width to 4-bit and then vary the weight bit-widths from 8 to 4. For the ImageNet experiments of Table 2, we use equal bit-widths for both weights and activations, following [1]. In CIFAR-10 experiment, all methods seem to maintain the performance of the quantized model until 4-bit quantization. Regardless of target bit-widths, PSGD outperforms all other regularization methods. On the other hand, ImageNet experiment generally shows reasonable results until 6-bit but the accuracy drastically drops at 4-bit. PSGD targeting 8-bit and 6-bit marginally improves on all bits, yet also experiences drastic accuracy drop at 4-bit. In contrast, Gradient L1 ($\lambda = 0.05$) and PSGD @ W4 maintain the performance of the quantized models even at 4-bit. Comparing with the second best method Gradient L1 ($\lambda = 0.05$) [1], PSGD outperforms it at all bit-widths. At full precision (FP), 8-, 6- and 4-bit, the gap of performance between [1] and ours are about 4.2%, 3.9%, 1.5% and 8.1%, respectively. From Table 2, while the quantization noise may slightly degrade the accuracy in some cases, a general trend that using more bits leads to higher accuracy is demonstrated. Compared to other regularization methods, PSGD is able to maintain reasonable performance across all bits by constraining the distribution of the full precision weight to resemble that of the quantized weight. This quantization-friendliness is achieved by the appropriately designed scaling function. In addition, unlike [1], PSGD does not need additional overhead of computing double-backpropagation.

**Post-training methods**     Table 3 shows that OCS, state-of-the-art post-training method, has a drastic accuracy drop at 4-bit. For OCS, following the original paper, we chose the best clipping method

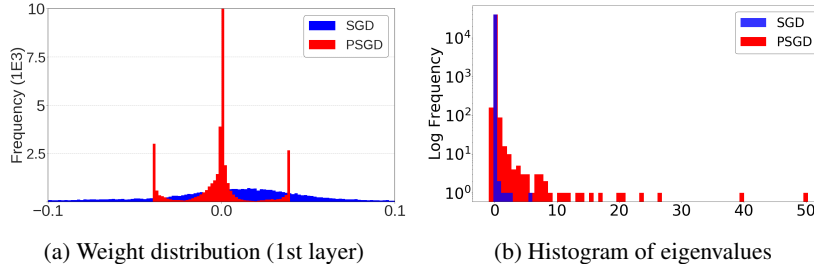

(a) Weight distribution (1st layer)  (b) Histogram of eigenvalues

Figure 4: Weight distribution and histogram of eigenvalues for MNIST dataset. The two-layered fully connected network consists of 50 and 20 hidden nodes. Target bit of PSGD is 2. Note that both solutions yield relatively small negative eigenvalues ($\lambda > -1$).

for both weights and activation. DFQ also has a similar tendency of showing drastic accuracy drop under the 6-bit as depicted in Fig. 1 of the original paper of DFQ [32]. This is due to the fundamental discrepancy between FP and quantized weight distributions as stated in Sec. 1 and Fig. 1. On the other hand, models trained with PSGD have similar full-precision and quantized weight distributions and hence low quantization error due to the scaling function. Our method outperforms OCS at 4-bit by around 19% without any post-training and weight clipping to treat the outliers. Applying PTQ method to our PSG pre-trained model is shown in Sec 5.

**Extremely low bits quantization**   As shown in Fig. 1, SGD suffers drastic accuracy drop at extremely low bits such as 3-bit and 2-bit. To confirm that PSGD can handle extremely low bit, we conduct experiments with PSGD targeting 3-bit and 2-bit except the first and last layers which are quantized at 8-bit. Table 4 shows the results of applying PSGD. Although the full precision accuracy does drop due to the strong constraints, PSGD is able to maintain reasonable accuracy. This demonstrates the potential of PSGD as a key solution to post-training quantization at extremely low bits.

## 5   Discussion

In this section, we first focus on the local minima found by PSG with a toy example to gain a deeper understanding. In this toy example, we train with SGD and PSGD on 2-bit on the MNIST dataset with a fully-connected network consisting of two hidden layers (50, 20 neurons). In the subsequent sections, we clarify the differences of the purpose of PSGD and that of QAT and analyze why PSGD does not achieve near-FP performance in LP. Lastly, we demonstrate the potential application of PSG by applying it to a concurrent PTQ method.

**Quantized and sparse model**   SGD generally yields a bell-shaped distribution of weights which is not adaptable for low bit quantization [41]. On the other hand, PSGD always provides a multi-modal distribution peaked at the quantized values. For this example, three target points are used (2-bit) so the weights are merged into three modes as depicted in Fig. 4a. A large proportion of the weights are near zero. Similarly, we note that the sparsity of ResNet-18@W4 shown in Table 2 is 72.4% at LP. This is because symmetric quantization also contains zero as the target point. PSGD has nearly the same accuracy with FP ($\sim$96%) at W2A32. However, the accuracy of SGD at W2A32 is about 9%, although the FP accuracy is 97%. This tendency is also shown in Fig. 1b, which demonstrates that PSGD reduces the quantization error.

**Curvature of PSGD solution**   In Sec 3.4 and Fig. 2, we claimed that PSG finds a minimum with sharp valleys that is more compression friendly, but has a less chance to be found. As the curvature in the direction of the Hessian eigenvector is determined by the corresponding eigenvalue [15], we compare the curvature of solutions yielded by SGD and PSGD by assessing the magnitude of the eigenvalues, similar to [5]. SGD provides minima with relatively wide valleys because it has many near-zero eigenvalues and the similar tendency is observed in [5]. However, the weights trained by PSGD have much more large positive eigenvalues, which means the solution lies in a relatively sharp valley compared to SGD. Specifically, the number of large eigenvalues ($\lambda > 10^{-3}$) in PSGD is 9 times more than that of SGD. From this toy example, we confirm that PSG helps to find the minima

which are more compression-friendly (Fig 4a) and lie in sharp valleys (Fig. 4b) hard to reach by vanilla SGD.

**Quantization-aware training vs PSGD**   Conventional QAT methods [12, 14] starts with a pre-trained model initially trained with SGD and further update the weights by only considering the low precision weights. In contrast, regularization methods such as our work and [1] starts from scratch and update the full-precision weights *analogous to SGD*. In our work, the sole purpose of PSGD is to find a set of full precision weights that are quantization-friendly so that versatile deployment as low precision (LP) is possible without further operation. Therefore, regularization methods start from the initial training phase analogous to SGD, whereas QAT methods starts with a pre-trained model after the initial training phase such as SGD and PSGD. The purpose of QAT methods is solely focused on LP weights. In general, a coarse gradient is used to update the weights attained by forwarding the LP weights, instead of the FP weights by using the straight-through-estimator (STE) [25]. Additionally, the quantization scheme is modified to include trainable parameters dependent on the low-precision weights and activations. Thus, QAT cannot maintain the performance of full-precision as it only focuses on that of low-precision such as 4 bit-width.

**Post-training with PSGD-trained model**   Our model attains similar full-precision performance with SGD and reasonable performance at low-precision even with naive quantization. Thus, PSGD-trained model can be potentially used as a pre-trained model for QAT or PTQ methods. We performed additional experiments using the model trained with PSGD in Table 3 by applying a concurrent PTQ work, LAPQ [33], using the official code. This attains 66.5% accuracy for W4A4, which is more than 3.1% and 6.2% points higher than that of PSGD-only and LAPQ-only respectively. This shows that PTQ methods can benefit from using our pretrained model.

## 6   Conclusion

In this work, we introduce the position-based scaled gradient (PSG) which scales the gradient proportional to the distance between the current weight and the corresponding target point. We prove the stochastic PSG descent (PSGD) is equivalent to applying the SGD in the warped space. Based on the hypothesis that DNN has many local minima with similar performance on the test set, PSGD is able to find a compression-friendly minimum that is hard to reach by other optimizers. PSGD can be a key solution to low bit post training quantization, because PSGD reduces the quantization error bridging the discrepancy between the distributions of the compressed and uncompressed weights. Because target points act as a prior to constrain original weights to be merged at specific positions, PSGD also can be used for sparse training by simply changing the target point as 0. In our experiments, we verify PSGD in the domain of pruning and quantization by showing the effectiveness on various image classification datasets such as CIFAR-10/100 and ImageNet. Also, we empirically show that PSGD finds minima which are located in sharper valleys than SGD. We believe that PSGD will help further researches in model quantization and pruning.

**Broader Impact**

PSG is a fundamental method of scaling each gradient component differently depending on the position of a weight vector. This technique can replace conventional gradient in any applications that require different treatment of specific locations in the parameter space. As shown in the paper, the easiest conceivable applications would be quantization and pruning where a definite preference for specific weight forms exists. These model compression techniques are at the heart of the fast and lightweight deployment of any deep learning algorithms and thus, PSG can make a huge impact in the related industry. As another potentially related research topic, PSG has a chance to be utilized in the optimization area such as the integer programming and the combinatorial optimization acting as a tool in optimizing a continuous surrogate of an objective function in a discrete space.

**Acknowledgments**

This work was supported by IITP grant funded by the Korea government (MSIT) (No.2019-0-01367, Babymind) and Promising-Pioneering Researcher Program through Seoul National University (SNU).

## Footnotes

[3]Details on $c(\bar{x})$ can be found in Appendix B. Also, another example of warping function and its experimental results are included in the same section.

[4]We set $\eta = \eta_0 \lambda_s$ where $\eta_0$ is the conventional learning rate and $\lambda_s$ is a hyper-parameter that can be set differently for various scaling functions depending on their range.

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
