[Supplementary Material]

# Position-based Scaled Gradient for Model Quantization and Pruning – Appendix

**Jangho Kim**
Seoul National University
Seoul, Korea
kjh91@snu.ac.kr

**KiYoon Yoo**
Seoul National University
Seoul, Korea
961230@snu.ac.kr

**Nojun Kwak**[*]
Seoul National University
Seoul, Korea
nojunk@snu.ac.kr

## A    Detailed Results on CIFAR-100 with ResNet-32 (Fig. 1)

In Table A1, we show the classification accuracies and the corresponding mean squared error (MSE) of PSGD depicted in Fig. 1 of the original paper. Also, the weight distributions of the model at various layers are shown in Fig. A1. In this experiment, we only quantize the weights, not the activations, to compare the performance degradation as weight bit-width decreases. The mean squared errors (MSE) of the weights across different bit-widths are also reported. The MSE is computed by the squared mean of the differences in full-precision weights and the low-precision weights across layers. As some variance in performance was observed for lower bit-widths, we report the mean±standard deviation for the 2-bit and the 3-bit experiments. As stated, PSGD successfully merges the weights to the target points and obtains quite low MSE until 3-bit and the 2-bit MSE of PSGD is more than 2 times smaller than that of conventional SGD.

In Fig. A1, we display the full-precision weight distributions of the PSGD models and compare them against vanilla SGD-trained distributions. Four random layers of each model are shown column-wise. The first row displays the model trained with SGD and L2 weight decay. Below are distributions trained with PSGD with target points for the sparse training case, 2-bit, 3-bit, and 4-bit respectively. Note that all the histograms are plotted in the full-precision domain, rather than the low-precision domain. For 2-bit, all three target bins ($2^2 - 1$) are visible. For 3-bit, only five target bins are visible as the peripheral two bins contain relatively low numbers of weight components.

Table A1: 8-, 6-, 4-, 3- and 2-bit weight quantization results of ResNet-32 learned by PSGD on the CIFAR-100 dataset. This is also reported in Figure 1 of the original paper. For lower bit-widths, the mean and standard deviation over five runs are reported. MSE between the learned weight and the target weight is reported. The numbers in the parentheses are the MSEs of SGD.

| Bit-width | 8-bit | 6-bit | 4-bit | 3-bit | 2-bit |
|---|---|---|---|---|---|
| Full precision | 70.14 | 70.59 | 70.08 | 68.96±0.34 | 67.16±0.40 |
| Low precision | 70.05 | 70.33 | 69.57 | 68.56±0.19 | 60.76±2.18 |
| MSE (SGD MSE) | 0.03 (0.03) | 0.41 (0.58) | 0.5 (11) | 0.84 (48) | 67 (157) |

## B    Methods

### B.1    Offset $c(\bar{x})$

In our warping function in the following

$$f(x) = 2\,\text{sign}(x - \bar{x})(\sqrt{|x - \bar{x}| + \epsilon} - \sqrt{\epsilon}) + c(\bar{x}), \tag{A1}$$

[*]Corresponding Author

Figure A1: Weight distributions in the full-precision domain of four random layers for sparse training, 2-bits, 3-bits, and 4-bits. The name of the layer and the number of parameters in parenthesis are shown in the column. The y-axis and x-axis of PSGD distributions are clipped appropriately for visualization purposes and the number of bins is all set to 100 for both PSGD and SGD.

we introduced $c(\bar{x})$ for making $f(x)$ continuous. If we do not add a constant $c(\bar{x})$, the $f(x)$ has points of discontinuity at every $\{(n + 0.5)\Delta | n \in \mathbb{Z}\}$ as depicted in Fig. A2, where $\Delta$ represents step size and $n\Delta$ means $n$-th quantized value identical to $\bar{x}$ corresponding to $x$. We can calculate the left sided limit and right sided limit at $n\Delta + 0.5\Delta$ using Eq. A1.

$$f(n\Delta + 0.5\Delta^-) = 2\left(\sqrt{0.5\Delta + \epsilon} - \sqrt{\epsilon}\right) + c(n\Delta) \tag{A2}$$

$$f(n\Delta + 0.5\Delta^+) = -2\left(\sqrt{0.5\Delta + \epsilon} - \sqrt{\epsilon}\right) + c((n+1)\Delta) \tag{A3}$$

Based on the condition that the left sided limit and the right sided limit should be the same (Eq. A2 = Eq. A3), we can get the following recurrence relation:

$$c((n+1)\Delta) - c(n\Delta) = 4\left(\sqrt{0.5\Delta + \epsilon} - \sqrt{\epsilon}\right). \tag{A4}$$

f(x) with c(x̄) (Green) and without c(x̄) (Red) across various step sizes (Δ)

$\Delta = 0.5$　　　　　　　$\Delta = 1$　　　　　　　$\Delta = 1.5$

Figure A2: Scaling function $f(x)$ for different step size $\Delta$. The red graph depicts $f(x)$ without $c(\bar{x})$ and the green graph depicts $f(x)$ with $c(\bar{x})$ (Eq. A5). Without $c(\bar{x})$, there are points of discontinuity at every $\{(n+0.5)\Delta | n \in \mathbb{Z}\}$. After adding $c(\bar{x})$ to the scaling function $f(x)$, it becomes a continuous function (green).

Using the successive substitution for calculating $c(\bar{x})$, it becomes

$$\cancel{c(\Delta)} - c(0) = 4\left(\sqrt{0.5\Delta + \epsilon} - \sqrt{\epsilon}\right)$$
$$\cancel{c(2\Delta)} - \cancel{c(\Delta)} = 4\left(\sqrt{0.5\Delta + \epsilon} - \sqrt{\epsilon}\right)$$
$$\vdots$$
$$+\ c(n\Delta) - \cancel{c((n-1)\Delta)} = 4\left(\sqrt{0.5\Delta + \epsilon} - \sqrt{\epsilon}\right)$$
$$\overline{\phantom{c(n\Delta) - c(0) = 4n\left(\sqrt{0.5\Delta + \epsilon} - \sqrt{\epsilon}\right)}}$$
$$c(n\Delta) - c(0) = 4n\left(\sqrt{0.5\Delta + \epsilon} - \sqrt{\epsilon}\right).$$

Setting $c(0) = 0$ and because $n\Delta = \bar{x}$, $c(\bar{x})$ can be calculated as below:

$$c(\bar{x}) = \frac{4\bar{x}}{\Delta}\left(\sqrt{0.5\Delta + \epsilon} - \sqrt{\epsilon}\right). \tag{A5}$$

## B.2 Non-separable directional scaling

Here, we introduce another example of warping function and the corresponding scaling function. In this case, we define the warping function as a multivariate function as $\mathcal{F}(\boldsymbol{x}) = [f_1(\boldsymbol{x}), \cdots, f_n(\boldsymbol{x})]^T$ and set

$$f_i(\boldsymbol{x}) = 2\,\text{sign}(x_i - \bar{x}_i)\sqrt{(\|\boldsymbol{x} - \bar{\boldsymbol{x}}(\boldsymbol{x})\|_\infty + \epsilon)\cdot(|x_i - \bar{x}_i| + \epsilon)} + c(\bar{x}_i). \tag{A6}$$

Here, $\|\boldsymbol{x}\|_\infty$ is the infinite norm or max norm which can be replaced with $|x_{max}|$ where $max$ is the index with the maximum absolute value. $c$ is a constant as in Eq. A1. By using $\delta_i = x_i - \bar{x}_i$, the partial derivative of Eq. A6 becomes

$$\frac{\partial f_i}{\partial x_j} = \begin{cases} \sqrt{|\delta_{max}| + \epsilon}/\sqrt{|\delta_i| + \epsilon} & \text{if } j = i \text{ and } j \neq max \\ 2 & \text{if } j = i \text{ and } j = max \\ \text{sign}(\delta_i)\text{sign}(\delta_j)\sqrt{|\delta_i| + \epsilon}/\sqrt{|\delta_j| + \epsilon} & \text{if } j \neq i \text{ and } j = max \\ 0 & \text{otherwise.} \end{cases} \tag{A7}$$

By changing the order of variable index, we can put the max element to the last and then the Jacobian matrix $J_x^{\mathcal{F}}$ becomes upper triangular with all-positive diagonal elements and the only non-zero off-diagonal elements are in the last column of the matrix. Comparing the magnitude of non-zero off-diagonal elements, which is in the range of $[-1, 1]$, with that of diagonal elements which is in the range of $[1, \sqrt{q/2\epsilon + 1}]$ where $q$ is the size of a quantization grid, off-diagonal elements does not dominate the diagonal elements. Furthermore, considering the deep network with a huge number

Figure A3: The scaling function $s(x_i)$ for each component in a 2-D weight space ($\boldsymbol{x} = (x, y)$ and $(0,0)$ is the target point). The independent scaling function in Eq.6 of the main paper (1st row) makes all the scaling factors $s(x_i)$'s for each gradient component very small when the weight vector $\boldsymbol{x}$ is very close to the target vector $\bar{\boldsymbol{x}}$. In the case of directional scaling (2nd row), in Eq. A6, even when the error between the target point and the weight vector is small, at least one direction, which corresponds to the element with the maximum error, has scaling of 1, possibly resulting in more stable learning.

of weight parameters, we can neglect the effect of off-diagonal elements and use only the diagonal elements of the Jacobian matrix for scaling. In this case, the elementwise scaling function becomes

$$s(x) = \frac{|x - \bar{x}| + \epsilon}{\|\boldsymbol{x} - \bar{\boldsymbol{x}}\|_\infty + \epsilon}. \tag{A8}$$

Using the elementwise scaling function Eq.(A8), the elementwise weight update rule for the PSG descent (PSGD) becomes

$$x_i^{t+1} = x_i^t - \eta s(x_i) \left. \frac{\partial \mathcal{L}}{\partial x_i} \right|_{\boldsymbol{x}^t}. \tag{A9}$$

**Independent vs Directional scaling:** The independent scaling function such as the one presented in the main paper (Eq.6) only considers the independent element-wise distance between the positions of weights and the targets. This means when the weight vector is very close to one of the target points, the magnitude of gradients could be very small, leading to slow convergence as the scaling function for all elements will be nearly 0. To avoid this, we added a small $\epsilon$ in Eq.6. Note, however, that the weights are needed to be updated according to the task loss (e.g. cross-entropy loss) to find an optimal solution. To address this degradation of gradient magnitude, directional scaling function (Eq.(A8)) finds the dominant direction by normalizing the scaled gradient as depicted in Figure A3. The directional scaling performs slightly better than the independent scaling as shown in Table A2, but the difference is not much. Note however that the vanishing of scaling function at the target in the independent scaling can be mitigated by increasing the offset $\epsilon$ in any way.

Table A2: 2-bit results on ResNet-18 with 90 epochs. Weight decay is not applied.

| Method | FP | W2A8 |
|---|---|---|
| Independent scaling | 63.35 | 54.96 |
| Directional scaling | 63.18 | 56.60 |

## C   Implementation details

We use CIFAR-10/100 and the ImageNet datasets for experiments. CIFAR-10 consists of 50,000 training images and 10,000 test images, consisting of 10 classes with 6000 images per class. CIFAR-100 consists of 100 classes with 600 images per class. The ImageNet dataset consists of 1.2 million images. We use 50,000 validation images for the test, which are not included in training samples. We use the conventional data pre-processing steps[2] [3].

**ImageNet / CIFAR-10**    For ResNet-18, we started training with a L2 weight decay of $10^{-4}$ and learning rate of 0.1, then decayed the learning rate with a factor of 0.1 at every 30 epochs. Training was terminated at 90 epochs. We only used the last 15 epochs for training the model with PSGD similar to [1]. This means we applied the PSG method after 75 epochs with learning rate 0.001. For extremely low-bits experiments, we did not use any weight decay after 75 epochs (See below). We tuned the hyper-parameters $\lambda_s$ for target bit-widths. All numbers are results of the last epoch. We used the official code of [8] for comparisons with 0.02 for the Expand Ratio[4].

**CIFAR-100**    For ResNet-32, the same weight decay and initial learning rate were used as above and the learning rate was decayed at 82 and 123 epoch following [7]. Training was terminated at 150 epoch. For VGG16 with batchnorm normalization (VGG16-bn), we decayed the learning rate at 145 epoch instead. We applied PSG after the first learning rate decay. The first convolutional layer and the last linear layer are quantized at 8-bit for the 2-bit and the 3-bit experiments. For sparse training, training was terminated at 200 epoch and weight decay was not used at higher sparsity ratio, while all the other training hyperparameters were the same. For [5], we used the official implementation for the results [5].

**Extremely low-bits experiments**    For ImageNet, we did not use the weight decay for 2-, 3-bits as it hinders convergence. For CIFAR100, weight decay was not used for only 2-bits. See the details regarding how weight decay affects training with PSGD in Sec. D.3. In addition, we experimented with training for longer epochs than the original schedule. In this case, we run additional 30 epochs for PSGD. The total number of epochs is 120 and we apply PSG methods for the last 45 epochs.

## D   Additional experiments

### D.1   Adam optimizer with PSG

To show the applicability of our PSG to other types of optimizers, we applied our PSG to the Adam optimizer by using the same scaling function with ResNet-32 on 4-bits with the CIFAR-100 dataset. Following the convention, the initial learning rate of $10^{-3}$ was used and the first and the last layer of the model were fixed to 8-bits. All the other training hyperparameters remained the same. Table A3 compares the quantization results of models trained with vanilla Adam and applying PSG to Adam.

Table A3: ResNet-32 trained with Adam on the CIFAR-100 dataset. Vanilla Adam also suffers accuracy degradation on 4 bits, while applying PSG to Adam recovers the accuracy by more than 5%. Weight-only quantization is shown by W4A32.

| Method | FP | W4A32 | W4A4 |
|---|---|---|---|
| Adam | 66.66 | 55.27 | 43.5 |
| Adam with PSG | 66.80 | 60.35 | 51.55 |

### D.2   Various architectures with PSGD

In this section, we show the results of applying PSGD to various architectures. Table A4 shows the quantization results of VGG16 [6] with batch normalization on the CIFAR-100 dataset and DenseNet-121 [3] on the ImageNet dataset, respectively.

(a) Conv2_2 layer          (b) Conv4_1 layer

Figure A4: The weight distribution of PSGD with weight decay and without weight decay.

For DenseNet, we run additional 15 epochs from the pre-trained model to reduce the training time [6]. For fair comparisons in terms of the number of epochs, we also trained for additional 15 epochs for SGD with the same last learning rate (0.001). However, we only observed oscillation in the performance during the additional epochs. Similar to the extremely low-bits experiments, we fixed the activation bit-width to 8-bit.

For VGG16 on the CIFAR-100 dataset, similar tendency in performance was observed with ResNet-32. The 4-bit targeted model was able to maintain its full-precision accuracy, while the model targeting lower bit-widths had some accuracy degradation.

Table A4: The performances of various architectures with PSGD.

| DataSet & Network | Method | (FP / W4A4) | (FP / W3A8) | (FP / W2A8) |
|---|---|---|---|---|
| CIFAR-100 & VGG16-bn | SGD | 73.12 / 63.08 | 73.12 / 3.44 | 73.12 / 1.00 |
| | PSGD | 73.21 / 70.92 | 71.85 / 68.28 | 69.36 / 53.25 |
| DataSet & Network | Method | (FP / W8A8) | (FP / W6A8) | (FP / W4A8) |
| ImageNet & DeseNet-121 | SGD | 74.43 / 73.85 | 74.43 / 70.57 | 74.43 / 0.36 |
| | PSGD | 75.16 / 75.03 | 75.12 / 74.84 | 72.60 / 72.26 |

### D.3 Weight decay at extremely low-bits

To show the weight decay effect on extremely low-bits with PSG, we trained models with and without weight decay with 90 epochs consisting of 75 epochs with SGD and last 15 epochs with PSGD. The results are shown in Table A5. Based on the experiment results, we found that weight decay incurred a detrimental effect on extremely low-bit cases (2,3-bit). Figure A4 shows the weight distribution of both models with and without weight decay. The range of the weight distribution with weight decay is smaller (Blue) than that of the weights without weight decay (Red) due to the weight shrinkage effect. This regularization effect does not matter at higher bit-widths such as 6-bit and 4-bit. However, it has a negative effect on the performance of extremely low-bits so we do not use weight decay for the extremely low-bits experiment.

Table A5: 6-,4-,3- and 2-bit results of ResNet-18 on the ImageNet dataset. All results are from training for 90 epochs.

| Method | (FP / W6A6) | (FP / W4A4) | (FP / W3A8) | (FP / W2A8) |
|---|---|---|---|---|
| PSGD with weight decay | 70.07 / 69.51 | 68.18 / 63.45 | 66.52 / 63.76 | 63.17 / 51.53 |
| PSGD without weight decay | 70.26 / 69.69 | 68.03 / 63.38 | 66.81 / 64.71 | 63.35 / 54.96 |

Figure A5: Visualizing the loss spaces of Fig. 5 using [4]; Left: Loss space of SGD solution; Right: Loss space of PSGD solution.

## D.4 Longer training for extremely low-bits

Although the model without weight decay does increase the performance significantly compared to the baseline, the performance gain is relatively lower in higher bit-widths (Table A5). We train for additional 30 epochs for PSGD and show the numbers in Table A6. In the table, we can see a significant performance enhancement by the longer training with PSGD. Note that additional training is not useful for bit-widths over 3-bit.

Table A6: ResNet-18 on the ImageNet dataset. The numbers in the second row are the results of longer training (120 epochs), which use additional 30 epochs with PSGD.

| Method | Weight decay | | No Weight decay | |
| --- | --- | --- | --- | --- |
| | (FP / W3A8) | (FP / W2A8) | (FP / W3A8) | (FP / W2A8) |
| PSGD | 66.52 / 63.76 | 63.17 / 51.53 | 66.81 / 64.71 | 63.35 / 54.96 |
| PSGD with additional epochs | 66.64 / 65.23 | 63.90 / 54.45 | 66.75 / 66.36 | 64.60 / 62.65 |

## E  Visualization of loss space

we have also used official code[7] of [4] to qualitatively assess the curvature of Fig. 5 in original paper, using the same experimental setting of Sec. 5, which is depicted in Fig A5 and it shows a similar tendency. The solution of PSGD is in the more sharp valley than it of SGD.

## F  Convergence analysis

Our algorithm is a variant of GD; the equivalent convergence analysis can be applied with the condition that the step-size, $0 < t \leq \frac{1}{L}$ where $L$ is a Lipschitz constant. The detailed definition and proof are in [2].

---
**Theorem:** Given the scaling vector, $s(\cdot) \in \mathbb{R}^n$ and a convex, L-smooth function, $f : \mathbb{R}^n \to \mathbb{R}$ satisfies: $f(x_{i+1}) - f(x_i) \leq \left(\frac{s(x_i)^{\circ 2} - 2s(x_i)}{2L}\right)^{\mathsf{T}} \nabla f(x_i)^{\circ 2}$, which is monotonically nonincreasing because r.h.s is always negative. As $i \to \infty$, $f(x_{i+1})$ converges to the optimum. ($\circ$ denotes the Hadamard operation)

**Proof:** Substituting $x_{i+1}$ for $x_i - \frac{s(x_i)}{L} \circ \nabla f(x_i)$ into r.h.s of $f(x_{i+1}) - f(x_i) \leq \nabla f(x_i)^{\mathsf{T}}(x_{i+1} - x_i) + \frac{L}{2}\|x_{i+1} - x_i\|_2^2$, (which follows from the property of L-smoothness) yields the inequality. Given $s(x_i) = \frac{abs(x_i - \bar{x}_i) + \epsilon}{\|x_i - \bar{x}_i\|_\infty + \epsilon}$ (Appendix B), which satisfies $0 < s(x_i)_j \leq 1, \forall j \in [1, n]$ the r.h.s is always negative. ($abs(\cdot)$ is defined as the element-wise absolute value function).

---

## G  Hyper-parameter $\lambda_s$

We searched the appropriate $\lambda_s$ with the criteria that the performance of the uncompressed model is not degraded, similar to [1]. For hyper-parameter tuning, we use two disjoint subsets of the training

dataset for training and validation. Then we used the found $\lambda_s$ to retrain on the whole training dataset. The below table shows the values of $\lambda_s$ used in experiments of the original paper. The $\lambda_s$ tended to rise for lower target bit-widths or for higher sparsity ratios (See Table A7 and A8). In CIFAR-10, we observe that same $\lambda_s$ value yields fair performance across all bit-widths.

Table A7: $\lambda_s$ used in the sparse training experiment.

| CIFAR-100 & ResNet-32 | Sparsity (%) | | | | |
| --- | --- | --- | --- | --- | --- |
| | 20.0 | 50.0 | 70.0 | 80.0 | 90.0 |
| $\lambda_s$ | 100 | 100 | 200 | 600 | 1200 |

Table A8: $\lambda_s$ used in the quantization experiments.

| ResNet-18 | ImageNet | | | CIFAR-10 | | |
| --- | --- | --- | --- | --- | --- | --- |
| | 8-bit | 6-bit | 4-bit | 8-bit | 6-bit | 4-bit |
| $\lambda_s$ | 500 | 500 | 1000 | 10 | 10 | 10 |

## Footnotes

[2] https://github.com/kuangliu/pytorch-cifar

[3] https://github.com/pytorch/examples/blob/master/imagenet/main.py

[4] https://github.com/cornell-zhang/dnn-quant-ocs

[5] https://github.com/AMLab-Amsterdam/L0_regularization

[6]`https://download.pytorch.org/models/densenet121-a639ec97.pth`

[7]`https://github.com/tomgoldstein/loss-landscape`