[Reviews · NeurIPS 2020]

Review 1

Summary and Contributions: The authors of this paper propose a methods to compress models. Specifically they propose that when learning parameters with gradient descent, the gradient should be rescaled to achieve sparse models. Rescaling is a volume preserving operation. Applying the scaled gradient is equivalent to warping the entire space. The authors evaluate their results, comparing to other quantisation schemes.

Strengths: I like the idea, it's simple and very effective. Double bonus for open source code. The paper is well organised.

Weaknesses: Figure 2 and the related section in the text is somewhat unclear to me. If the authors could elaborate and think about a more intuitive figure I think this would benefit the quality of the paper a lot.

Correctness: I don't see any issues.

Clarity: I like the organisation of the paper. This will not influence my judgement but I would recommend the authors to have the English checked. The text is hard to read here and there, especially wrt the use of “the”/“a”.

Relation to Prior Work: There have been some recent studies [A], [B]. That have also approached quantisation. I think the authors could compare/ reference them. (I apologise if these studies are too recent they might be, at least the authors could name them as concurrent work.) [A] van Baalen, Mart, Christos Louizos, Markus Nagel, Rana Ali Amjad, Ying Wang, Tijmen Blankevoort, and Max Welling. "Bayesian Bits: Unifying Quantization and Pruning." arXiv preprint arXiv:2005.07093 (2020). [B] Bhalgat, Yash, Jinwon Lee, Markus Nagel, Tijmen Blankevoort, and Nojun Kwak. "LSQ+: Improving low-bit quantization through learnable offsets and better initialization." In Proceedings of the IEEE/CVF Conference on Computer Vision and Pattern Recognition Workshops, pp. 696-697. 2020.

Reproducibility: Yes

Additional Feedback: Figure 4(a) and 3 could be scaled with log on the y-axis.


Review 2

Summary and Contributions: The paper proposes a new quantization aware training by changing the optimization algorithm. The new method (PSGD) is scaled the weights parameters that can be compressed with minimal accuracy loss at the inference phase. Specifically, the position of weights is changed so that the MSE between the weight and quantized weights is reduced even at 32-bit. The PSGD is compared with previous approaches such as SGD and L1 gradient. It has shown improvement in terms of performance over the other learning algorithms.

Strengths: 1- The proposed quantization aware training is evaluated with the previous method and outperforms the previous approaches in terms of performance. 2- The idea of finding the weight space that considers both quantization and performance simultaneously is novel and interesting.

Weaknesses: 1- It would be suggested this approach is also compared with reference [1,2] to improve the paper quality. 2- It would be recommended to evaluate the PSGD where the pruning and quantization are combined. [1] De Sa, Christopher, et al. "High-accuracy low-precision training." arXiv preprint arXiv:1803.03383 (2018). [2] Tambe, Thierry, et al. "AdaptivFloat: A Floating-Point Based Data Type for Resilient Deep Learning Inference." arXiv preprint arXiv:1909.13271 (2019).

Correctness: Yes

Clarity: The paper is well written, and the proof is clearly explained.

Relation to Prior Work: yes

Reproducibility: Yes

Additional Feedback: ==== Update after author feedback ==== The author addressed most of my review comments. However, more analysis is needed to explain why this method could not achieve similar accuracy with some previous approaches, such as LSQ and DSQ.


Review 3

Summary and Contributions: This paper proposed a regulation method, adding a scaling factor proportional to the distance between the precise and quantized values of weights in the weight update function. By doing so, the authors claim better accuracies in the benchmarks than the previous methods in both pruning and quantization.

Strengths: The method seems to work on quantization as well as pruning by setting the quantized value as zero for pruning.

Weaknesses: The theoretical proof and empritical demonstration are relatively weak. The theoretical proof has no convergence analysis, but just an abstract illustration based on Tayler expansion. The empirical methods are not clear to the reviewer. The authors claimed that their method is advanced compared to other methods requiring training dataset (line 38), but also mentioned that their method follows the training-from-start practice(line 53). The appendix C seems to show that the training dataset is still demanded. In Eq. 7, weights are updated based on Loss, is this loss just the training loss calculated from the training data? The reviewer cannot find the definition of this loss in the paper. The empirical demonstration (mainly on Fig.2) also didn't show the realistic case while the network is multidimensional and deep, for example, using loss landscape technique to demonstrate the advantage of the method.(Li, Hao, et al. "Visualizing the loss landscape of neural nets." Advances in Neural Information Processing Systems. 2018.) Moreover, the work failed to include this work(Banner, Ron, Yury Nahshan, and Daniel Soudry. "Post training 4-bit quantization of convolutional networks for rapid-deployment." Advances in Neural Information Processing Systems. 2019), which has better results than this method for the re-training-free quantization endeavors. If the work actually still requires training data(and/or label), the bar is even higher.

Correctness: The claims need to be verified with stronger theoretical analysis and more realistic empirical methods. And the claims that the method is better than other methods requiring training data(line 38) is confusing as the method also needs re-training according to the author(line53).

Clarity: The paper is fairly written.

Relation to Prior Work: No. The method didn't classify itself well into one of the two big categories: (1) post-trainig with training data(and/or label) and (2) methods requires no training data.

Reproducibility: Yes

Additional Feedback: After rebuttal comments: Thanks for the response from the authors. The authors confirmed that the method is a "regularization" method requiring training(and training data). In that case, I don't think the ~5% degradation on W4A4 ResNet18 inference justifies the advantages claimed. The performance of PSGD is significantly worse than other quantization-aware training(QAT) work. Although the authors refer to their method as a regularization method, essentially it requires similar training data and computation as QAT. In contrast, some post-training-quantization(PTQ) does not need training data but mainly tune the FP32 model with statistical method or distillation, which has a significant advantage at deployment if they can achieve FP32 accuracy. If the PTQ requires training data, then it falls back to the same level as QAT. Since PSGD requires training from scratch, not requiring an FP32 model does not seem to be an advantage to me---with training data/resources, one can always obtain an FP32 model. For QAT, W4A4 already shows no degradation in most of SOTA models. For example, LSQ method shows no degradation until W3A3, compared to ~5% degradation in this work at W4A4(Esser, Steven K., et al. "LEARNED STEP SIZE QUANTIZATION." ICLR. 2019). DSQ method achieves no degradation on W4A4 and ~4% degradation on W2A2(Gong, Ruihao, et al. "Differentiable soft quantization: Bridging full-precision and low-bit neural networks." ICCV. 2019. I skipped many papers on arxiv or published this year, which are competing for the W2A2 inference with the smallest accuracy loss. My understanding is that if training is allowed(tuning or training with regularization), 4b inference is already robust on Resnets, as shown by Nvidia's MLPerf result with less than 1% degradation on ResNet50 using INT4 hardware, https://developer.nvidia.com/blog/int4-for-ai-inference. If training data is not allowed, 4b inference without degradation is still a milestone to reach. Unfortunately, in the rebuttal, the authors confirm that it is not the case. In the rebuttal, the authors combined their method with PTQ (instead of QAT), and yield ~3% degradation from baseline for W4A4. However, I think it would be more meaningful to combine their method with QAT instead of PTQ. Because QAT and regularization both need training hardware and data, their combination induces no additional cost. In contrast, combining PTQ and regularization method actually weakens PTQ, which was valuable than QAT exactly because it doesn't require training hardware/data. Although the idea of PSGD is potential and interesting, some advantage over SOTA results is required to stand out among numerous work on quantization. Therefore, I will maintain my score on this work.


Review 4

Summary and Contributions: This paper proposes a method for quantizing the parameters of a neural network. The main idea is to regularize the weights so that they are aligned to a grid that makes them more quantization-friendly.

Strengths: The addressed problem is relevant and timely. Altough the idea is not completely novel, there is enough novetly in my opinion to justify a publication. The article is well organized and understandable, at least for someone experienced in the field. The experimental section provides credible results even if the benchmark references could be more recent. It was appreciated the attempt to provide some theoretical background to the proposed technique.

Weaknesses: The way (1) is derivated in 3.1 is not completely clear: more details should be provided in the following proof. It is not clear why teh function (5) wa selected: could the authors provide a better rationale for this choice ? In the experimental section, the comparison is with weak techniques or few-shot (like SNIP) techniques, making the comparison not very relevant. The authors shall consider more recent references. Fig. 4 is troubling: apparently, SGD finds a very flat spot while PSGD finds a solution in a non-flat region: the authors shall elaborate more on why the solution found by PSGD would be a better solution. The writing could be improved: the quality of English should be improved, if needed resorting to a proofreading service.

Correctness: Claims and methods are correct, even tough it should be better eplained how (1) is derivated.

Clarity: It is understandable and clear, however the level of English shall be improved.

Relation to Prior Work: References to prior work are adequate, even tough stronger experimental references are needed.

Reproducibility: Yes

Additional Feedback:

[Author Response · NeurIPS 2020]

We thank the reviewers for their positive and constructive feedbacks. We tried our best to respond to all the raised issues and will reflect them in the final version.

**[S1] PSGD is a Regularization method:** Post-training quantization (PTQ) methods need a pre-trained model. However, a model pre-trained with SGD suffers from the problem shown in Fig. 1 (line 44-49). To tackle this issue, we train a compression-friendly model at full-precision (FP) with cross-entropy loss using PSGD. Our method can be considered as a regularization method equivalent to [1] (line 50-60, 80-89). After training, we can use simple layer-wise quantization to obtain a low-precision (LP) model without any data nor post-training (line 257), while PTQ methods need calibration data or additional computing phases. Note that our PSGD has a similar accuracy with the SGD-trained model at FP.

**[S2] Comparison to other Post-training methods:** We employ layer-wise quantization. ACIQ [2] uses channel-wise quantization (e.g. scale factor and zero point *per channel*) which attains higher performance at the expense of hardware-friendliness as noted in many prior works [7, 26, 34]. We had already cited the work and included the differences between the two methods in Sec. 2 (line 72-79). A similar rationale is given in Sec. 5.1 of a concurrent

Figure 5: Loss surface using [35]; SGD (top) and PSGD (bottom)

work [34] for not comparing with channel-wise methods. In Table 1 of ACIQ [2], the naive (channel-wise) baseline of ResNet-18 W4A4 (ImageNet) is 51.6% as opposed to 0.3% for ours (layer-wise). Hence, improving layer-wise quantization is a much more challenging problem that deserves attention because of its hardware efficiency. We have already compared with SOTA layer-wise methods in Table 2&3. Additionally, our PSGD can be combined with PTQ methods because we do not use any post-training. We performed additional experiments using a model trained with PSGD then post-processing with a concurrent PTQ work, LAPQ [34], using the official code. This attains 66.5% accuracy for W4A4, which is more than 3.1% and 6.2% points higher than that of PSGD-only and LAPQ-only respectively. Note that at lower bits such as W2A8, we attain 62.7% accuracy, while LAPQ has 1.3% accuracy.

**[S3] Convergence analysis:** Our algorithm is a variant of GD; the equivalent convergence analysis can be applied with the condition that the step-size, $0 < t \leq \frac{1}{L}$ where $L$ is a Lipschitz constant. The detailed definition and proof are in [38].

---

**Theorem:** Given the scaling vector, $s(\cdot) \in \mathbb{R}^n$ and a convex, L-smooth function, $f : \mathbb{R}^n \to \mathbb{R}$ satisfies: $f(x_{i+1}) - f(x_i) \leq (\frac{s(x_i)^{\circ 2} - 2s(x_i)}{2L})^{\mathsf{T}} \nabla f(x_i)^{\circ 2}$, which is monotonically nonincreasing because r.h.s is always negative. As $i \to \infty$, $f(x_{i+1})$ converges to the optimum. ($\circ$ denotes the Hadamard operation)

**Proof:** Substituting $x_{i+1}$ for $x_i - \frac{s(x_i)}{L} \circ \nabla f(x_i)$ into r.h.s of $f(x_{i+1}) - f(x_i) \leq \nabla f(x_i)^{\mathsf{T}}(x_{i+1} - x_i) + \frac{L}{2} \|x_{i+1} - x_i\|_2^2$, (which follows from the property of L-smoothness) yields the inequality. Given $s(x_i) = \frac{abs(x_i - \bar{x}_i) + \epsilon}{\|x_i - \bar{x}_i\|_\infty + \epsilon}$ (Appendix B), which satisfies $0 < s(x_i)_j \leq 1, \forall j \in [1, n]$ the r.h.s is always negative. ($abs(\cdot)$ is defined as the element-wise absolute value function).

---

**Reviewer 1 (R1):** Thank you for the positive feedback! We will include the suggested recent studies, revise the figures, and check with the English in the final version. Regarding Fig. 2, we will do our best to intuitively present our idea.

**Reviewer 2 (R2):** We are pleased the reviewer pinpointed the keypoints of the paper. *I. Suggestion for comparison:* These papers [36,37] propose to use LP arithmetic at the training phase or use different representation format to encode the parameters. While direct comparison may be difficult, we believe that the idea proposed by [37] can be incorporated into our pre-trained model for future work. An additional experiment applying a PTQ method is presented in [S2]. *II. Evaluation with pruning:* As pointed out, PSGD also achieves high sparsity as zero is included in the target set. The sparsity of ResNet-18@W4 (ImageNet) at LP is 72.4%! We will reflect *I* and *II* in the final version.

**Reviewer 3 (R3):** Thank you for the meaningful feedback. *I. Convergence analysis:* Please refer to [S3] for the convergence analysis. *II. Position of our method:* We apologize if we caused any confusion. Our method is not a post-training method and further details are in [S1]. *III. Visualization of loss space:* We respectfully disagree to the claim that empirical demonstration was not shown in a realistic case, as Sec. 5 and Fig. 4(b) compare the curvature of the solutions of a neural network. As suggested, we have also used official code of [35] to qualitatively assess the curvature in Fig. 5, using the same experimental setting of Sec. 5, which shows a similar tendency. *IV. Stronger Baseline:* Suggested baseline, ACIQ [2] is a channel wise quantization method. Detailed explanation regarding why channel-wise method was not compared is in [S2]. We will reflect all issues to avoid any confusion.

**Reviewer 4 (R4):** We are glad that the reviewer apprehended our novelty. *I. Taylor Expansion:* The equation is derived using Taylor expansion around $\mathbf{y}_t$ for the first equality. $\mathcal{F}^{-1}(\mathbf{y}_t - \eta \nabla_{\mathbf{y}}^{\mathcal{L}'}(\mathbf{y}_t)) = \mathcal{F}^{-1}(\mathbf{y}_t) + \mathcal{J}_{\mathbf{y}}^{\mathbf{x}}(\mathbf{y}_t)(\mathbf{y}_t - \eta \nabla_{\mathbf{y}}^{\mathcal{L}'}(\mathbf{y}_t) - \mathbf{y}_t) = \mathcal{F}^{-1}(\mathbf{y}_t) - \eta \mathcal{J}_{\mathbf{y}}^{\mathbf{x}}(\mathbf{y}_t) \nabla_{\mathbf{y}}^{\mathcal{L}'}(\mathbf{y}_t)$. *II. Eq.(5):* Eq.(5) is derived by selecting the scaling factor (Eq.(6)), $s(x) = \frac{1}{[f'(x)]^2}$, which is the scale multiplied to the gradient of the original space. Then, Eq.(5) can be interpreted as the warping function. The motive for Eq.(6) is explained in (line 137-142). *III. Pruning baseline:* We only considered single-shot pruning [22,25] because the intention of the experiment was to see the effectiveness of PSGD on making weights converge to zero (line 215-220). Comparing recent pruning methods and applying iterative pruning schedules to PSGD is our future work which is not the scope of this work. *IV. Fig. 4:* The intention of this section was to point out that PSGD solution cannot be found by standard SGD as it lies in a much sharper local minimum. The validity of the PSGD solution is explained in Sec. 3.4. and [S3]. Moreover, the solution is more quantization-friendly than that of SGD because it reduces the quantization error (refer to Fig.1 and Sec. 3.2). We will reflect raised issues for clearer understanding.

[34] Nahshan, Yury, et al. "Loss Aware Post-training Quantization." arXiv preprint arXiv:1911.07190 (2019).
[35] Li, Hao, et al. "Visualizing the loss landscape of neural nets." Advances in Neural Information Processing Systems. 2018.
[36] De Sa, Christopher, et al. "High-accuracy low-precision training." arXiv preprint arXiv:1803.03383 (2018).
[37] Tambe, Thierry, et al. "AdaptivFloat: A Floating-Point Based Data Type for Resilient Deep Learning Inference." arXiv preprint arXiv:1909.13271 (2019).
[38] EE236C, L. Vandenberghe. "1. Gradient method."


[Meta-Review · NeurIPS 2020]

The authors propose an interesting technique for quantizing by rescaling gradients during training. It is an interesting idea that can built upon even if the results are not SOTA. The reviewers have concerns that the comparisons in the paper are not appropriately made with other QAT approaches. It is recommended that the authors address this to make their paper as strong as possible.